# Knowledge and Use of the 2011 Freedom of Information Act among Journalists in Nigeria

**Ogemdi Uchenna Eze** 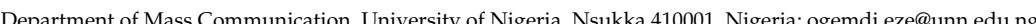

Department of Mass Communication, University of Nigeria, Nsukka 410001, Nigeria; ogemdi.eze@unn.edu.ng

**Abstract:** This study examined the knowledge and use of the 2011 Freedom of Information Act among journalists in Nigeria. The hierarchy of influences model provided the theoretical lens, which guided the study. Through a survey of 313 Nigerian journalists, the study found that there was a high level of knowledge of the Act among Nigerian journalists. Nigerian journalists perceived the Act as a useful journalistic tool, and they often used it for such purposes as confirming facts, writing controversial topics and to gain insight into the inner working of government. The study showed that, in the use of the Act in journalistic duties, Nigerian journalists were confronted with the challenges of non-integration of the provisions of the Act in the operations of government agencies, adversarial disposition of government institutions towards journalists and the pervasive culture of secrecy. The study established that knowledge of the Act positively correlated with its use. The implications of the findings were discussed.

**Keywords:** FOI Act; journalists; Nigeria; information; hierarchy of influences model





## 1. Introduction

On 28 May 2011, Nigeria achieved a milestone in its quest towards entrenching democratic structures, which will ensure good governance, when the then President Goodluck Jonathan signed the Freedom of Information (FOI) bill into law. The Act, among other things, seeks to "make public records and information more freely available, provide for public access to public records and information, protect public records and information to the extent consistent with the public interest" (Dunu and Ugbo 2014, p. 1). By this action, Nigeria joined the league of a few African nations, such as Ghana, Malawi, Senegal, Uganda, the Democratic Republic of Congo, Cameroon, Ethiopia, South Africa, Madagascar and Tanzania, which have clear and specific guarantees of the right to information entrenched in their constitution (Ojo 2010). The push towards the introduction of FOI laws around the globe is underpinned by growing evidence, which supports the notion that there is a direct relationship between the level of freedom the press exercises and the level of corruption in society (Brunetti and Weder 2003; Solis and Antenangeli 2017). A free press is one of the most potent antidotes to bureaucratic corruption (Solis and Antenangeli 2017). The current global movement towards improved access to government-held information is premised on the longstanding notion that freedom of information brings about transparency, which curtails corruption (Ogunniran 2016). Writing within the context of commonwealth nations, Paterson and McDonagh (2017) describe the continued spread of FOI laws as a welcome development, stressing that it demonstrates the acknowledgement of the import of transparency, both in strengthening democracy and discouraging financial misapplication and graft in governments. Scholars have established a relationship between poverty and corruption. Corruption inhibits growth (Mauro 1995), whereas corruption itself negatively impacts income and education (Glaeser and Saks 2004). It is argued that confronting bureaucratic corruption could result in improved service delivery programme, which will ultimately reduce poverty (Francken et al. 2009).

While there are statutory institutions saddled with the responsibility of fighting corruption, Francken et al. (2009) argue that they may not be effective, as these institutions seem to be equally corrupt, particularly in countries with a high prevalence rate of capture of public funds. A more effective approach could be strengthening the media, through the FOI Act, to carry out their monitorial role. The performance of the press' watchdog role is contingent on the degree to which the government is willing to permit media scrutiny (Márquez-Ramírez et al. 2020; Kalogeropoulos et al. 2022). A key test for measuring such willingness is the introduction of FOI law. With the passage of the FOI Act in Nigeria, there is a compelling need for an appraisal of journalists' use of the FOI Act in their journalism practice.

Several reasons render this study imperative. First of all, the news media in most young democracies are struggling to assert their independence from state control (Freedom House 2014). As a result, investigating the process, which enables or undermines this process, is pertinent because of the critical role of independent media in democracy and development (Camaj 2015). Secondly, the proponents of transparency see FOI laws as an advantage with teleological expectations, which spread to other socio-democratic rights for the wider citizenry (Calland and Bentley 2013). Since journalists are more likely to use FOI laws than individual citizens (Smolina 2005), it is important to investigate how the FOI Act has affected journalism practice in Nigeria. Thirdly, the majority of FOI adopters are found in developing nations with limited resources, largely unstable political environments, as well as weaker administrative and legal systems than the earlier adopters in the West (Roberts 2006). It is noted that the uptake in FOI laws' adoption around the world coincided with the third wave of democratisation (Kato and Tanaka 2019), which took place in Africa. Some studies have shown that adoption of the FOI Act in some countries was performed as a ploy to bolster their democratic credentials (Michener 2011; Paterson 2008). Most studies have focused on legal analysis, examining the interpretation and functionality of FOI laws (Paterson 2005; Snell 2004), while their practical effects on journalism have not been given much attention in the developing world, particularly in Nigeria. Fourthly, early studies conducted on the FOI Act and journalism in Nigeria paid greater attention to journalists' perception of the Act in terms of its workability, how it will affect journalism practice and its capacity to enhance journalism practice (Dunu and Ugbo 2014), with little effort being devoted to investigating its actual use by journalists. This is why the present study builds on this line of research by investigating journalists' knowledge of the Act, journalists' perception of the Act as a useful journalistic tool, the circumstances and the extent to which Nigerian journalists use the Act in their journalism practice, as well as the challenges faced by Nigerian journalists in the application of the FOI Act in journalism practice. Again, since some of these studies were carried out shortly after the passage of the Act, such assessment may be premature, given that journalists, the media and the government may still be coming to terms with the new FOI Act environment. Therefore, studying how journalists use the Act in their profession more than a decade after the passage of the bill into law presents ample opportunity to make a valid judgement regarding FOI Act usage among Nigerian journalists. In what follows, we discuss the FOI Act and the news media, outline the theoretical frameworks guiding the study and review the relevant literature. We outline the methodology adopted in the study, present our findings, discuss them and conclude the study.

## 2. FOI Act and News Media

Normatively, the FOI Act was mainly fashioned by journalists, for journalists and with a specific intention that journalists would utilise the access to government information in order to provide knowledge for the citizenry, which will, by extension, facilitate citizens' effective participation in the democratic process (Kwoka 2013). Although some scholarly works have pointed to the centrality of journalists both to FOI Act enactment and its expected use (Silver 2016; Camaj 2015; Paterson 2008; Lidberg 2002), the available evidence indicates that global FOI laws have not been able to meet the expectations of the news

media (Dunu and Ugbo 2014; Snell 1998; Fajans 1984). Silver (2016) admits that the most notable criticism of the FOI Act lies in its inability to meet the information needs of the news media. The reasons for failure of the FOI Act to meet the news media's information needs are to be found in two entities: the law itself and its handlers.

Concerning the law itself, some FOI laws can be counterproductive. Michener (2011) observes that FOI laws in Zimbabwe and Paraguay were so twisted that they ended up limiting access to information previously available and restricting freedom of expression. Even when the law does not limit free expression, some provisions of the FOI Act constitute hindrances to the efficient use of the Act. First, there are time limits within which the information required can be supplied (Paterson 2008). This is at variance with the journalistic norm of immediacy (Deuze 2005). Hence, the law does not lend itself to some journalistic types, except investigative reports, where journalists will have ample time to wait for long. Non-investigative journalists may be inclined to seek information through options, which give faster responses than FOI requests (Hayes and Silke 2019; Cuillier 2010). Second, in some FOI laws, and in particular in Nigeria's FOI Act, there are charges to be borne by the requester. Paterson (2008) notes that the cost of access is a deterrence to media usage of FOI law. Silver (2016) states that economic considerations may affect FOI Act requests by members of the news media. Rydell (2011) shares the same view, arguing that with declining revenue in the media, journalists' FOI Act requests and contests for FOI refusal may be waning. For Kreimer (2008), challenging denials in courts is not only costly and time consuming but puts pressure on resources, which are in short supply in the media. The third issue related to application of the FOI Act among journalists is that some FOI laws have contentious categories of exemptions, which protect deliberative processes in government agencies (Paterson 2008). Such exemptions, Paterson (2008, p. 8) explains, are critical for journalists in the performance of their watchdog role, as these documents are needed to shine a light on the "thinking processes", which underpin government decision making. Furthermore, they illuminate what particular advice was being followed in reaching an important policy decision.

*Government Secrecy*

In considering the socio-political factors affecting the use of FOI law by journalists, it is pertinent to observe that "governments everywhere have a natural inclination towards secrecy" (Paterson 2008, p. 3). Although the rapid spread of FOI laws has made enactment of the Act fashionable for nation states, the evidence suggests that in some countries, there is little or no political or bureaucratic capacity to implement the Act. The fact that FOI law has been enacted, Michener (2011) explains, makes a political leader appear good, and the international community feels pleased, notwithstanding the legal quality of what was passed. Consequently, Michener (2011) cautions that, regardless of FOI laws' merits and the prevailing drive for transparency, many encumbrances could render these laws ineffectual. First, there is less likelihood that states whose enactment of FOI law is predicated on political exigency will muster the political will required to drive its implementation. Calland and Bentley (2013) observe that several studies on the effectiveness of different FOI Acts share common characteristics of impediments, which include political will, infrastructural inadequacy, deficits in records management, procedural defects hindering responsiveness and institutional culture of secrecy. With regard to the culture of secrecy, there exists a tension between governments—the bureaucracy in particular—and the news media in the use of the FOI Act in journalism practice. The two are opposed to each other, with one interested in concealing information and the other being focused on exposing it. Michener and Worthy (2015) explain that the motivation of the FOI requester triggers two *prima facie* motivations—that is, to probe the government or to be informed—as a pre-political act. They conclude that FOI laws are a public service, which assumes government responsibility, which seems to prompt defensive speculation on the part of government officials. Such reaction is to be expected, and rightly so, when the identities of FOI scholars and advocates are considered (Michener and Worthy 2015). The authors note that political narratives

appear to dominate other interpretations of FOI use. This defensive speculation is further heightened when the requesters are identified as journalists or media organisations. The implementation of FOI law is a capital-intensive venture, which requires both monetary investment and training of staff to handle such requests (Obayi et al. 2020). Regrettably, many countries stop their journey to open government at the point of FOI passage, thereby missing the potential benefits of the Act. The extent to which this can be said of Nigeria has not been empirically verified; hence, this study investigates the knowledge and use of the FOI Act among journalists in Nigeria.

## 3. The Hierarchy of Influences Model

The present study uses the hierarchy of influences model to examine Nigerian journalists' knowledge and use of the FOI Act in their journalism duties. This model is particularly useful, in that it explores the factors enabling as well as inhibiting journalists' knowledge and use of the Act by shinning a light on five levels of influences on journalistic content, namely the individual, routine, organisational, extra-media and social institution (formerly known as ideological) levels.

### 3.1. Individual Level

The individual level relates to the biological, psychological and sociological characteristics of an individual social actor. Here, the attitudes, training and background of the journalist (or media worker more generally) are seen as influential (Shoemaker and Reese 1996). Empirical evidence viewed the demographic characteristics of journalists as the sources of individual-level influences (Relly et al. 2015).

### 3.2. Routine Level

The routine level involves the patterned, recurring practices and guidelines, which journalists apply in the performance of their work. Reese and Shoemaker (2016, p. 399) elucidate that "the routines level is concerned with those patterns of behaviour that form the immediate structures of media work . . . including unstated rules and ritualized enactments that are not always made explicit". Hanitzsch et al. (2010) and Hanitzsch and Mellado (2011) argue that policies, conventions and customs wield strong influence on journalistic practices globally. However, in some cases, journalists become used to problematic behaviour. For example, Relly et al. (2015) found that most journalists in Ba'athist Iraq did not have any ethical issues with regard to paying a source for information.

### 3.3. Organisational Level

The organisational level refers to the policies, rules and economic considerations within media organisations, which affect media content. Here, the profit orientation shared by private media companies, combined with their hierarchical structure, in general, shape the content in line with the ownership's interests (Hackett and Uzelman 2003). Editorial policy, Shoemaker and Reese (1996) explain, allows the organisation to determine which stories are deemed newsworthy, how they are prioritised and how they are represented.

### 3.4. Extra-Media Level

The extra-media level relates to the norms, individuals and organisations, which function outside a given media outlet. This represents the meso-level environment for the media, that is, the interrelationships of economic, political and cultural factors lying between the organisation and the society in general. At the extra-media level, consideration is given to those influences emanating principally from outside the media organisation. This perspective recognises that the power to shape content is not entirely that of the media, but it is shared with a variety of institutions in the society, including the government, advertisers, public relations, influential news sources, interest groups and even other media organisations (Shoemaker and Reese 1996).

*3.5. Social Institution Level*

The ideological level refers to the symbolic frameworks or customs, ideals and theories, which exist at the societal level. Ideology not only shapes the news; it is extended, renewed and reproduced through the agency of media content. In this context, we are interested in how the media's symbolic content is related to larger social interests and how meaning is mobilised in the service of power. This necessarily leads us to consider how each of the previous levels operates in order to add up to a coherent ideological result (Shoemaker and Reese 1996).

Although the model was developed to account for patterns of media coverage, it has been adapted to the study of journalists' attitudes and perceptions (Berkowitz and Limor 2003; Weaver and Wilhoit 1996). The hierarchy of influences model is relevant to this research, as it provides a useful analytical framework through which journalists' knowledge and use of the FOI Act in journalism practice can be interrogated. This is so because the model moves from the individual to the social institution level, which recognises both the agency (individual) and the structure (organisational and extra-media) in the use of the FOI Act. For instance, at the individual level, we are interested in understanding how journalists' knowledge and perception of the Act affect their use of the legislation. In terms of routine, the type of job a journalist performs can influence their knowledge and use of the FOI Act. For instance, investigative journalists may use the Act to greater effect than non-investigative journalists. We may be interested in understanding how the news routine of reporting events as quickly as possible affects the use of the FOI Act by journalists. In terms of the organisational level, variations in the editorial policy concerning the FOI Act in a media house can have a far-reaching effect on how journalists use the Act in their work. For example, a supportive media organisation may inspire increased use of the Act and vice versa. At the extra-media level, journalists who meet bureaucrats who are resistant or uncooperative while trying to access government-held information may not be willing to use the Act in the future, or if the culture of secrecy is pervasive, this can result in discontinuation of the use of the Act or exploration of other means of accessing government-held information, e.g., tipping government officials for such documents. At the social institution level, the prevailing belief can impact how journalists use the Act in their work. For instance, if the influence of the transparency movement leading to enactment of the FOI Act within a society was borne out of a desire to launder a regime's image, the state may not provide all the tools, which enable smooth implementation of the FOI Act. This can affect the way the Act is used by journalists in a particular locale. Some studies have shown that adoption of the FOI Act in some countries was carried out as a ploy to bolster their democratic credentials (Michener 2011; Paterson 2008).

## 4. FOI Act in Journalism Practice

*4.1. Knowledge of the FOI Act among Journalists*

Since the passage of the FOI bill into law in 2011, various stakeholders have made efforts at educating Nigerians, particularly journalists, on the provisions of the Act. Non-governmental organisations—the Media Rights Agenda (MRA) and International Press Centre (IPC)—have consistently trained journalists on how to use the Act (Dunu and Ugbo 2014; Adeyemoye 2020). It is suggested that knowledge of the Act will facilitate its use. Knowledge is identified as a major first stage in the process of adopting an innovation (Vejlgaard 2018). Within the context of the use of the FOI Act in journalism practice, knowledge of the Act among journalists has been of particular interest among journalism scholars.

Apuke (2017), who examined the level of knowledge of the provisions of the FOI Act among journalists in Jalingo, Taraba State, found that most journalists had an impressive level of knowledge of the Act. A study by Egielewa and Aidonojie (2021) revealed that there was poor knowledge of the Act among the journalists in Edo State. For their part, Obayi et al.'s (2020) study used the contents of the Act to test the journalists' knowledge level and found that there was a high level of knowledge of the provisions of the FOI Act

among journalists in Imo State. For their part, Nnadi and Obot's (2014) study showed that 65.1% of journalists in Akwa Ibom State have not read the FOI Act and, by extension, were not knowledgeable on the provisions of the Act. This result should be interpreted in light of the fact that the study was conducted three years after the passage of the FOI Act. It is logical to expect more journalists to have read the Act over time.

### 4.2. Journalists' Perception of the FOI Act

Scholars (Vu et al. 2017; Hanitzsch and Vos 2017; Van Dalen et al. 2011) have established a connection between journalists' perceptions and actions. For instance, Vu et al.'s (2017) study found that journalists in Vietnam perceived male sources as being more agentic than female sources. A content analysis of the stories written by these journalists reflected their perception of their sources, in that male sources were depicted as possessing stronger work-oriented and agentic traits, whereas their female counterparts were framed as being more socially oriented and communal. A large body of knowledge on role conception is premised on the notion that the way in which journalists perceive their role affects how they perform it (Mellado et al. 2020; Hanitzsch and Vos 2017). In line with this view, examining how journalists perceive the FOI Act becomes a legitimate enterprise to investigate.

Egielewa and Aidonojie (2021) found that most respondents perceived the Act as capable of facilitating easier retrieval of information from public institutions. Obayi et al. (2020) investigated the perception of the FOI Act among journalists in Imo State. The result of the study showed that journalists in Imo State had a positive perception of the Act. Uche (2017) assessed the perception of the FOI Act among journalists in Awka, Anambra State, and found that the respondents perceived the Act as capable of making journalism practice easier, as well as guaranteeing a free and responsible press in Nigeria. Similarly, Aliyu's (2017) study showed that journalists perceived the Act as a tool, which will advance professionalism in journalism. Abone and Kur (2014) found that 77.6% of the journalists in Anambra State perceived the Act as having the potential to improve journalism practice in the country, However, in their study of journalists' perception and use of the FOI Act, Ezinwa et al. (2014) found that more than 70% of the respondents disagreed with the idea that the FOI Act is an important legal document, which could support investigative reporting in the country.

### 4.3. The Use of the FOI Act by Journalists

One of the arguments made in favour of the passage of the FOI Act into law was that it would enhance the practice of journalism in the country. In fact, some experts averred that it would revolutionise the journalism practice in Nigeria. As Ahmad et al. (2022, p. 2) argue, "Laws on their own do not bring change until they are used". Consequently, some studies have investigated the degree of use.

Arguably, one of the earliest comparative studies assessing the degree of use of the FOI Act within the journalism industry could be traced to Waters' (1999) study, which indicated a relatively low-level use of the FOI Act among Australian journalists compared to their American counterparts. Similarly, Lamble (2004) compared media use of FOI law among US, Canada, New Zealand and Australian journalists. The result showed that New Zealand journalists were the heaviest users of FOI law in the English-speaking world. Additionally, New Zealand journalists made more lodgements than their US counterparts. However, the study showed that US journalists used FOI law to better effect than the rest. The author attributed frequent use of the FOI Act in journalism practice in New Zealand to the conducive FOI regime in operation in the country.

In Nigeria, a 2018 study by Agba, Ogri and Adomi showed that journalists in Calabar were yet to use the Act to hold public officials accountable to the people. In a ten-year study of the use of the FOI Act in journalism practice in Nigeria, Asogwa et al. (2021) found that there was low use of the FOI Act in journalism practice in the country. A related study by Nnadi and Obot (2014) found that just 22.3% of the journalists included in the study had ever filed an application for information based on the provisions of the Act. Among

the number that applied, 78.4% of the applications were denied. Hence, it was concluded that there was poor utilisation of the FOI Act in journalistic practice. Gamji and Abdul (2019) investigated the influence of the FOI Act on journalistic practice and found, among other things, that the use of the FOI Act by journalists in their profession was low due to subsisting anti-press laws.

*4.4. Challenges*

In the use of the FOI Act in their profession, journalists face many challenges. This fact has maintained the interest of scholars working in the transparency field. A study by Abone and Kur (2014) identified the continued existence of other anti-press laws—such as the Official Secret Act, Penal Code and Criminal Code—the cost and time spent on litigation in the event of denial, the absence of a supervisory body, a culture of secrecy and disregard for the law as challenges to utilization of the FOI Act in journalism practice. Similarly, Agba et al. (2018) identified ignorance on the part of requesters, denial of access to public information by public officers, executive immunity and rigid legal procedures as challenges militating against the use of the FOI Act in journalism practice. Exploring journalists' access to and constraints in applying the FOI Act in Rivers State, Ituma et al. (2019) found that government policies topped the factors, which impeded journalists' access to information. Other constraints included fear of being killed, the policy of the media organisation and poor record keeping in Nigeria. Madubuike-Ekwe and Mbadugha (2018) found non-compliance by government or public officials with provisions of the Act, a pervasive culture of secrecy among government institutions, poor record keeping and non-domestication of the FOI Act as impediments to implementation of the FOI Act. Silver (2016) found delays and processing inefficiencies related to access requests as major reasons for the law failing journalists.

In light of the above, the following research questions formed the thrust of the study:

1. What is the level of knowledge of the FOI Act among Nigerian journalists?
2. How much do Nigerian journalists perceive the FOI Act as a useful journalistic tool?
3. Under what circumstances and to what extent do Nigerian journalists use the Act in their journalism practice?
4. What are the challenges faced by Nigerian journalists in the application of the FOI Act in journalism practice?
5. What is the relationship between journalists' use of the FOI Act and their knowledge of the Act?

## 5. Methods

In order to answer the central question of this research, the study undertook a national survey of Nigerian journalists from 15 October to 15 December 2021. The study used the multi-stage sampling technique. First, Nigerian journalists were stratified into six geo-political zones. Nigeria consists of six geo-political zones, and in order to achieve representativeness of journalists in Nigeria, respondents needed to be drawn from every geo-political zone. In the second stage, the researchers used the purposive sampling technique to select one chapter of the Nigeria Union of Journalists (NUJ) from each of the six geo-political zones because these states are media hubs in their respective zones. In the third stage, four corresponding chapels (the smallest unit of the NUJ, which is located in a media house) were selected using the random sampling technique. In the fourth stage, we employed the random sampling technique to administer copies of the questionnaire to journalists in the selected correspondent chapels.

The NUJ was used as the basis for sampling because it is a union, which houses all journalists in Nigeria and has a register from which a sample can be drawn. Hence, the population of the study totalled 4299 journalists, with the population of each chapter represented in the following order: Lagos State = 2236; Abuja = 1267; Enugu State = 192; Rivers State = 202; Kaduna State = 221; Sokoto State = 181 journalists. Using Wimmer and Dominick's online sample size calculator with a 95% confidence level and a 5% margin of

error, we obtained a sample size of 353 journalists. We used Bowley's sampling technique in allocating questionnaires to the selected chapters, as follows: Lagos State = 184; Abuja = 104; Kaduna State = 18; Enugu State = 16; Rivers State = 16; and Sokoto State = 15 journalists. Furthermore, the accidental sampling technique was used in the administration of the questionnaire to the respondents. A total of 313 questionnaires were completed and found useful for analysis, representing an 88.7% return rate.

The instrument for data collection was the questionnaire. The questionnaire comprised two sections: A—demographic section and B—psychographic section. All items in the psychographic section were measured using a four-point Likert scale, ranging from strongly disagree (1) to strongly agree (4). In order to validate the instrument, the questionnaire was subjected to face validity by research experts in the field of journalism. The test–retest reliability of the measurement instrument was assessed by administering the instrument to a sample of 60 members of the NUJ Anambra State chapter on two different occasions, with a two-week interval between these administrations. The scores obtained from both administrations were then compared using the interclass correlation coefficient (ICC). ICC (2, 1) was chosen, as it is suitable in situations where there is a single rater or where the focus is on the consistency of ratings across administrations. The ICC value was found to be 0.83, showing a high level of agreement and consistency between the two administrations. This suggests that the measurement instrument demonstrated good stability over time.

Data obtained from the field were sorted, coded and screened using the Statistical Package for the Social Sciences (SPSS) computer-based statistical software, version 26.0. The demographic variables were analysed using frequency and percentage. Mean and standard deviation were used to answer the research questions, while ANOVA was used for testing all null hypotheses at a 0.05 level of significance.

## 6. Results

The result of the study, as presented in Table 1, shows the demographic information of respondents by gender, age, educational qualification, marital status, length of service and the medium they work for. The result shows that the majority of respondents were male, representing 60.1%, while 39.9% were female respondents. The majority of respondents were between 30 and 39 years of age, had a HND/BSc qualification, were married, had spent 11–15 years in journalism and worked in a radio station.

**Table 1.** Demographic information of respondents.

| SN | Gender | Frequency | Percentage |
|----|--------|-----------|------------|
| 1 | Male | 188 | 60.1% |
| 2 | Female | 125 | 39.9% |
| | Total | 313 | 100% |

| SN | Age Range | Frequency | Percentage |
|----|-----------|-----------|------------|
| 1 | 20–29 Yrs | 25 | 8.0% |
| 2 | 30–39 Yrs | 69 | 22.0% |
| 3 | 40–49 Yrs | 111 | 35.5% |
| 4 | 50–59 Yrs | 79 | 25.2% |
| 5 | 60 and above | 29 | 9.3% |
| | Total | 313 | 100.0% |

| SN | Qualification | Frequency | Percentage |
|----|---------------|-----------|------------|
| 1 | OND | 27 | 8.6% |
| 2 | HND/BA/BSc | 179 | 57.2% |
| 3 | MA/MSc | 97 | 31.0% |
| 4 | PhD | 10 | 3.2% |
| | Total | 313 | 100.0% |

**Table 1.** *Cont.*

| SN | Marital Status | Frequency | Percentage |
|----|----------------|-----------|------------|
| 1 | Single | 70 | 22.4% |
| 2 | Married | 209 | 66.8% |
| 3 | Divorced | 15 | 4.8% |
| 4 | Widowed | 19 | 6.1% |
| | Total | 313 | 100.0% |

| SN | Yrs in Service | Frequency | Percentage |
|----|----------------|-----------|------------|
| 1 | 1–5 Yrs | 81 | 25.9% |
| 2 | 6–10 Yrs | 73 | 23.3% |
| 3 | 11–15 Yrs | 79 | 25.2% |
| 4 | 16–20 Yrs | 39 | 12.5% |
| 5 | 21 Yrs and above | 41 | 13.1% |
| | Total | 313 | 100% |

| SN | Medium of Work | Frequency | Percentage |
|----|----------------|-----------|------------|
| 1 | Television | 67 | 21.4% |
| 2 | Radio | 93 | 29.7% |
| 3 | Newspaper | 91 | 29.1% |
| 4 | Magazine | 24 | 7.7% |
| 5 | Online | 38 | 12.1% |
| | Total | 313 | 100.0% |

Ordinary National Diploma (OND); Higher National Diploma (HND).

- RQ 1: What is the level of knowledge of the FOI Act among journalists in Nigeria?

The finding of the study, as presented in Table 2, shows the frequency, mean and standard deviation of respondents regarding the level of knowledge of the FOI Act among journalists in Nigeria. The result shows that the majority of journalists have good knowledge of the FOI Act in Nigeria. This is because the respondents agreed with the following statements: "A public institution shall update and review the information required to be published under this section periodically, and immediately, whenever changes occur" ($\bar{x}$ = 3.23, SD = 0.89) and "Where a case of wrongful denial of access is established, the defaulting officer or institution commits an offence and is liable on conviction to a fine of N500,000" ($\bar{x}$ = 2.91, SD = 0.84), among others. This is because the mean ratings are above 2.50, which was set as a criterion for accepting an item. A cluster mean of 3.04 with a standard deviation of 0.85 implies that journalists in Nigeria have good knowledge of the FOI Act.

**Table 2.** Mean and standard deviation of the level of knowledge of the FOI Act among journalists in Nigeria.

| SN | Item Statement | Criteria | $\bar{x}$ | SD | Decision |
|----|----------------|----------|-----------|-----|----------|
| 1 | A public institution shall update and review the information required to be published under this section periodically, and immediately, whenever changes occur. | +2.5 | 3.23 | 0.89 | A |
| 2 | Where a case of wrongful denial of access is established, the defaulting officer or institution commits an offence and is liable on conviction to a fine of N500, 000. | +2.5 | 2.91 | 0.84 | A |
| 3 | A public institution may deny an application for any information the disclosure of which may be injurious to the conduct of international affairs and the defence of the Federal Republic of Nigeria. | +2.5 | 3.00 | 0.91 | A |
| 4 | A notification of denial of any application for information or records shall state the name, designation and signature of each person responsible for the denial of such application. | +2.5 | 3.02 | 0.79 | A |
| | Cluster Mean | +2.5 | 3.04 | 0.85 | A |

The criteria consist of a benchmark or cut-off point on the basis of which a statement is accepted or rejected.

- RQ 2: How much do Nigerian journalists perceive the FOI Act as a useful journalistic tool?

The finding of the study, as presented in Table 3, shows that journalists believe the FOI Act will improve journalism practice in Nigeria ($\bar{x}$ = 3.60, SD = 0.60), and they feel confident when using the FOI Act to carry out their journalistic duties ($\bar{x}$ = 3.38, SD = 0.70). However, journalists disagreed about the FOI Act being a useless piece of legislation, which will make the practice of journalism complex ($\bar{x}$ = 1.99, SD = 1.01) and about the Act diverting journalists' attention from the core issues affecting their profession ($\bar{x}$ = 2.10, SD = 1.00). A cluster mean of 2.77 with a standard deviation of 0.80 implies that journalists in Nigeria agreed that the FOI Act is a useful journalistic tool.

**Table 3.** Mean and standard deviation of the perception of the FOI Act among Nigerian journalists as a useful journalistic tool.

| SN | Item Statement | Criteria | $\bar{x}$ | SD | Decision |
|----|----------------|----------|-----------|-----|----------|
| 1 | I believe the FOI Act will improve journalism practice in Nigeria. | +2.5 | 3.60 | 0.60 | A |
| 2 | I feel confident when I use the FOI Act to carry out my journalistic duties. | +2.5 | 3.38 | 0.70 | A |
| 3 | I consider it a useless piece of legislation, which will make the practice of journalism complex. | +2.5 | 1.99 | 1.01 | D |
| 4 | It diverts journalists' attention from the core issues affecting our profession. | +2.5 | 2.10 | 1.00 | D |
| | Cluster Mean | +2.5 | 2.77 | 0.80 | A |

- RQ 3: Under what circumstances and to what extent do Nigerian journalists use the Act in their journalism practice?

The result in Table 4 shows that, to a high extent, journalists employ the Act whenever they want to write anything controversial ($\bar{x}$ = 2.56, SD = 1.03); journalists use the Act to gain access to documents exposing them to the inner workings of government ($\bar{x}$ = 2.73, SD = 0.97); and they also use the Act as a confirmatory mechanism ($\bar{x}$ = 2.79, SD = 0.92). However, the result shows that making FOI Act requests as a journalistic routine is low ($\bar{x}$ = 2.39, SD = 1.02). A cluster mean of 2.62 with a standard deviation of 0.94 implies that journalists in Nigeria use the FOI Act in their journalism practice to a high extent.

**Table 4.** Mean and standard deviation of respondents on the extent journalists use the FOI Act in their journalism practice.

| SN | Item Statement | Criteria | $\bar{x}$ | SD | Decision |
|----|----------------|----------|-----------|-----|----------|
| 1 | I employ the Act whenever I want to write anything, which is controversial. | +2.5 | 2.56 | 1.03 | HE |
| 2 | I use the Act to gain access to documents, which expose me to the inner workings of government. | +2.5 | 2.73 | 0.97 | HE |
| 3 | I use the Act as a confirmatory mechanism. | +2.5 | 2.79 | 0.92 | HE |
| 4 | Making FOI Act requests has become routine for me. | +2.5 | 2.39 | 1.02 | LE |
| | Cluster Mean | +2.5 | 2.62 | 0.94 | HE |

- RQ 4: What are the challenges faced by journalists in the application of the Act in their practice?

The result in Table 5 shows that the following are challenges faced by journalists in the application of the Act in their practice: Government institutions are yet to integrate the provisions of the FOI Act in their operations, as there are no information officers responsible for FOI requests, nor do they make certain information publicly available ($\bar{x}$ = 3.02, SD = 0.91); Some government agencies believe that the FOI Act does not apply to them, and so, they do not honour any requests based on the Act ($\bar{x}$ = 3.07, SD = 0.83); Some government agencies still adopt an adversarial disposition towards journalists making requests for information ($\bar{x}$ = 3.25, SD = 0.75); Obtaining information is difficult, as the culture of secrecy is still heavily entrenched in government business ($\bar{x}$ = 3.44, SD = 0.66).

A cluster mean of 3.20 with a standard deviation of 0.78 means that the respondents agreed that journalists in Nigeria have been facing some challenges in the application of the FOI Act in their practice.

**Table 5.** Mean and standard deviation of respondents on the challenges faced by journalists in the application of the Act in their practice.

| SN | Item Statement | Criteria | $\bar{x}$ | SD | Decision |
|---|---|---|---|---|---|
| 1 | Government institutions are yet to integrate the provisions of the FOI Act in their operations, as there are no information officers responsible for FOI requests, nor do they make certain information publicly available. | +2.5 | 3.02 | 0.91 | A |
| 2 | Some government agencies believe that the FOI Act does not apply to them, and so, they do not honour any request based on the Act. | +2.5 | 3.07 | 0.83 | A |
| 3 | Some government agencies still adopt an adversarial disposition towards journalists making requests for information. | +2.5 | 3.25 | 0.75 | A |
| 4 | Obtaining information is difficult, as the culture of secrecy is still heavily entrenched in government business. | +2.5 | 3.44 | 0.66 | A |
| | Cluster Mean | +2.5 | 3.20 | 0.78 | A |

- RQ 5: What is the relationship between journalists' use of the FOI Act and their knowledge of the Act?

Pearson's product correlation of knowledge of the Act and its use was found to be weakly positive and statistically significant. This shows that as journalists' knowledge of the Act increases, more journalists tend to use the Act in their work ($r = 0.278$, $p = 0.05$).

## 7. Discussion of Findings

This paper examined the knowledge and use of the FOI Act in journalism practice in Nigeria. This study is pertinent, given that the Act has been in use for more than a decade. The result of the study indicated that respondents had a high level of knowledge of the Act, in that the majority of them could correctly identify the provisions of the Act. This finding is consistent with previous studies. For instance, Apuke (2017) found that there was a high level of knowledge of provisions of the FOI Act among journalists in Taraba State. Similarly, Obayi et al.'s (2020) study showed that journalists in Imo State had a high level of knowledge of the Act. However, the result of the present study is at variance with a study by Egielewa and Aidonojie (2021), who found that there was poor knowledge of the Act among journalists in Edo State. Based on the hierarchy of influences model, an analysis of journalists' knowledge of the Act can be pursued on three levels, namely the individual, routine and organisational levels. At the individual level, a journalist who likes to be conversant with developments in their field may experience the need to become acquainted with the Act, knowing that such knowledge will help them operate efficiently within the journalistic field. In terms of influences based on routine, it is expected that a journalist who wants to engage in investigative reporting will be keen on acquiring knowledge of the Act in order to function well, as they are routinely seeking to uncover what some people want to be kept from the public (Suntai and Shem 2018; Pollack and Allern 2018). Thus, we contend that the demands of one's routine will either motivate or discourage one from acquiring knowledge of the Act. The interest shown by an investigative reporter in the FOI Act will not be the same as that of an entertainment journalist. Shoemaker and Reese (1996) note that media organisations wield particular influence on media content. In the context of this study, the editorial policy of a media organisation can encourage or discourage the acquisition of knowledge of the FOI Act among journalists. A medium, which prioritises the monitorial role of journalism, will encourage its journalists to read and study the FOI Act in order to function well in the area of holding the government to account.

Our finding showed that journalists perceived the FOI Act as a useful journalistic tool. Most journalists thought that the Act would improve the journalism practice in Nigeria and felt confident when they used the Act to carry out their journalistic duties. Again,

most journalists did not agree that the Act was a useless piece of legislation, which would make journalism practice complex, nor that the Act diverted journalists' attention from the core issues affecting their profession. This positive perception of the FOI Act as a useful journalistic tool mirrors existing studies. Abone and Kur (2014) found that nearly 80% of journalists in Anambra State perceived the Act as having the capacity to enhance journalism practice in the country. Aliyu's (2017) study showed that respondents perceived the Act as a tool, which will further professionalism in journalism. A study by Obayi et al. (2020) indicated that journalists in Imo State had a positive perception of the Act. However, the result of the present study runs contrary to a study by Ezinwa et al. (2014), which revealed that 70% of journalists did not perceive the Act as an important legal document, which could facilitate investigative reporting in Nigeria. Based on the hierarchy of influences model, an analysis of the perception of the FOI Act as a useful journalistic tool could be explored at two levels. At the individual level, a journalist's perception of the usefulness of the Act as a journalistic tool may be informed by their attitude towards the Act. This attitude is also shaped by the experiences of journalists attempting to use the Act. In a situation where they frequently and successfully make use of the Act in the discharge of their duties as a journalist, they will perceive the Act as useful, and vice versa. At the extra-media level, the disposition of government agencies towards the FOI regime could shape the perception of journalists regarding the utility of the Act in journalism practice. At this level, there is an assumption that the media operate in organised relationships with other institutions, which work to shape media content. A further supposition is made that these relationships can be coercive but are more often voluntary and collusive (Shoemaker and Reese 1996). In the event of a culture of secrecy, uncooperative attitude of government officials and unnecessary delays dominating the administration of the FOI Act, the perception of its usefulness may be dimmed on account of such behaviour.

The extent and the circumstances under which journalists use the FOI Act in their work—according to our finding—showed that respondents used the Act whenever they want to write anything controversial, to gain access to documents exposing them to the inner workings of government and as a confirmatory mechanism. However, the result showed that the routinisation of FOI-Act-enabled requests in journalism practice among Nigerian journalists has been achieved to a low extent. Therefore, the result indicated that, although journalists in Nigeria use the FOI Act in their journalism practice to a low extent, the circumstances under which they use the Act diverged along writing on controversial topics, as a confirmatory mechanism and as a way of gaining insight into government's operations. This result echoes previous studies (Asogwa et al. 2021; Gamji and Abdul 2019; Nnadi and Obot 2014; Dunu and Ugbo 2014; and Agba et al. 2018), which found a low use of the FOI Act in journalism in Nigeria. However, the current result is at variance with that of Chukwu and Ihejirika (2018), who found that journalists in Lagos State used the Act to access information to a high extent. Drawing from the hierarchy of influences model, we argue that the circumstances under which the FOI Act has been deployed in this study may be interrogated at two levels, namely the routine and ideological levels. At the routine level, journalists are socialised into the culture of verification. It is the responsibility of journalists to confirm that the information at his or her disposal is correct and verifiable. The FOI Act offers an opportunity to achieve that in the current study. Relatedly, delving into controversial topics demands that the journalist provides evidence to back up their claims, and this is what journalists in this study were doing. At the ideological level, although the essence of the FOI Act is to make government information publicly available, what was often encountered is that a request had to be made before information could be released. This speaks to the influence of the transparency movement in the adoption of the FOI Act. If the FOI Act is adopted in order to bolster the democratic credentials of a country, the effect is that journalists and other citizens interested in government-held information may have to request the information before it is made available. This may also manifest itself in the non-appointment of information officers for handling FOI Act requests.

The challenges faced by journalists in utilization of the FOI Act in their profession include non-integration of the provisions of the FOI Act in the operation of government institutions, government agencies' attitude that the Act does not apply to them—and, as a result, not honouring any requests based on the Act—adversarial disposition of government institutions towards journalists requesting information using the FOI Act and the difficulty in obtaining government-held information occasioned by a pervasive culture of secrecy in government business. This finding is supported by previous studies. For example, this study identified that the failure of government institutions to integrate the provisions of the FOI Act in their operations—as evidenced in no information officers being charged with FOI requests—coheres with the absence of a supervisory body charged with the responsibility of responding to FOI Act requests found in Abone and Kur's (2014) study. Adversarial disposition towards journalists making a request using the FOI Act was found to be common in some government agencies. This is in tandem with previous studies (Agba et al. 2018; Dunu and Ugbo 2014), which identified uncooperative disposition of government agencies as an impediment to application of the Act by journalists. This study also found that a culture of secrecy is still pervasive in government business. This result is consistent with previous studies (Madubuike-Ekwe and Mbadugha 2018; Obayi et al. 2020; Abone and Kur 2014), where a culture of secrecy in government institutions was identified as a challenge to utilization of the FOI Act in the discharge of journalistic duties. In the context of the hierarchy of influences approach, our result showed that the major challenge to the use of the FOI Act in journalism practice revolved around government and its agencies, which represents the extra-media level. An enabling environment, which will facilitate the FOI regime, such as the appointment of information officers saddled with the task of responding to requests, is largely non-existent. Some government agencies seem to arrogate to themselves the status of being not answerable to the FOI regime in Nigeria, and thus, they do not honour requests for documents made through the FOI Act. We argue that the adversarial disposition of government agencies towards journalists making FOI requests would adversely affect their capacity to maximise the benefits of the Act to journalism, just as a culture of secrecy would. In the event where journalists cannot access the information they require to fulfil their role in the society, they may altogether abandon the task of holding the powerful to account. As noted earlier, the FOI Act is mostly used as a confirmatory mechanism; in a situation where journalists cannot perform such confirmation, their reportage will be based on tips or unverified anonymous sources. The overall effect is that their reports could be easily discounted as unreliable stories. This problem may be surmounted by establishing relations with government officials, taking part in advocacy campaigns to champion the implementation of the FOI Act and fashioning legal strategies to challenge government agencies, which fail to comply with the FOI Act.

The present result showed that knowledge of the Act correlated with the use of the Act by journalists. This study highlights the importance of knowledge in facilitating the use of the FOI Act in journalism practice. It is little wonder that one of the participants in a study by Mohammed et al. (2023) questioned how he could use the Act when he does not know its content. It is for this reason that journalists have been trained consistently on how to use the Act (Dunu and Ugbo 2014; Adeyemoye 2020). Informed by the hierarchy of influences model, knowledge of the Act may be interrogated at the individual and extra-media levels. At the individual level, the attitude of the journalist plays a role in determining how well a journalist learns and understands the Act. A journalist who believes that the enactment of the Act was predicated on boosting the democratic credentials of a government may not pay much attention to studying the Act. At the extra-media level, some organisations, such as the Media Rights Agenda (MRA) and International Press Centre (IPC), believe that by training journalists on how to use the Act (that is, increasing their knowledge of the Act), they can encourage the use of the Act among journalists in Nigeria.

## 8. Conclusions

This paper examined the knowledge and use of the FOI Act among Nigerian journalists. Given the promise of revolutionising the journalism practice, which the Act was said to engender, it is important to assess the knowledge and use of the Act among journalists. A high level of knowledge of the Act among journalists in Nigeria is indicative of the fact that they are well informed of their right to access information and can potentially use it in the discharge of their work. The general positive perception of the Act among journalists shows that it is a valuable asset for journalists in Nigeria and can contribute to the enhancement of the journalism profession in the country. With the deployment of the FOI Act as a confirmatory mechanism, tips or information received from anonymous sources can be confirmed or refuted through the Act, thereby improving the journalism profession through accurate and reliable reportage of events. The use of the FOI Act has not become routine among journalists in Nigeria; instead, they use the Act in certain circumstances, e.g., as a confirmatory mechanism, to write controversial stories and to gain insight into the inner workings of government. The challenges journalists face in their use of the FOI Act in journalistic duties are many, but they essentially revolve around the adversarial disposition of government towards journalists requesting information. This can have serious implications for journalists' performance of their monitorial role.

**Funding:** The author received no financial support for the publication of this paper.

**Institutional Review Board Statement:** The study was conducted in accordance with the Declaration of Helsinki, and approved by the Faculty of Arts, University of Nigeria Nsukka Research Ethics Committee with protocol code UN/FAREC/02232021 and 15 June 2021.

**Informed Consent Statement:** Informed consent was obtained from all subjects involved in the study.

**Data Availability Statement:** Data will be made available on request.

**Conflicts of Interest:** The author declares no potential conflicts of interest with respect to the research, authorship and/or publication of this article.

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
