# Peer review of "Knowledge and Use of the 2011 Freedom of Information Act among Journalists in Nigeria"

_journalmedia, doi:10.3390/journalmedia5010017_

Round 1

Reviewer 1 Report

Comments and Suggestions for Authors

See review comments attached

Author Response

Response to Reviewer 1

Reviewer’s comment: Lines 47_52 need citation, otherwise they seem line opinion.

Response: Citation has been provided for lines 47-52.

Reviewer’s comment: Line 53 – elaborate on these struggles.

Response: I have elaborated on the struggles highlighted in line 53

Reviewer’s comment: Line 61– “Thirdly, the majority of FOI adopters are in developing nations with limited resources” (this to needs elaboration).

Response: I have elaborated on line 61

Reviewer’s comment: Line 67– Please contract reason No. 4 with this study’s purpose(s)

Response: I have contracted reason No 4 with the study’s purpose.

Reviewer’s comment: Line 116– This paragraph is better being under a separate subheading on government secrecy. This improves clarity and flow.

Response: I have put line 116 under the sub-heading “Government secrecy”.

Reviewer’s comment: Line 143–144 need citation.

Response: I have added citation to lines 143-144

Reviewer’s comment: The discussion on the hierarchy of influences model is not clear and is not clearly tied to FOI as well. The author first needs to tell what each of the five levels in the hierarchy are, then how each enhances this study.

Response: With regard to the theoretical framework, I have stated the five levels and how each enhances the study.

Reviewer’s comment: These three subsections under the lit. review sections seem disjointed from the rest of the paper.

Response: The literature review was done in conformity with the research questions. The first sub-section addressed knowledge, the second, perception, the third addressed use while the fourth looked at the challenges faced by journalists while using the Act in their work.

Reviewer’s comment: Why have a theoretical framework and a lit. review at the same time. These passages need to be incorporated in other parts of the paper or be under a different subheading that is closely tied to the study’s purpose(s). For instance, isn’t the use of the FOI Act by journalists more useful under the FOI Act and News Media section?

Response: There is no particular reason for having literature review and theoretical framework sections.

Reviewer’s comment: Explain why the six-region demarcation was appropriate.

Response: Nigeria is constitutive of six geo=political zones. To achieve representativeness, it is fitting to draw sample from these zones.

Reviewer’s comment: Likewise, explain why the NUJ was used a basis for sampling.

Response: The NUJ is the union to which almost all Nigerian journalists belong to  and it has a register from which a sample can be drawn.

Reviewer’s comment: For audiences outside of Nigeria, explain what chapel means.

Response: The chapel is the smallest unit of the NUJ located within a media house.

Reviewer’s comment: In table one, provide a key at the bottom for the terms OND and HND.

Response: I have provided a key at the bottom for the terms OND and HND.

Reviewer’s comment: Please explain what criteria means.

Response: I pointed out that criteria is the benchmark on which an item can be accepted or rejected.

Reviewer’s comment: We also do not need the SA, A, and SD when we have the means. This creates confusion

Response: I have deleted the SA, A, D and SD

Reviewer’s comment: Also explain what Dec means.

Response: Dec is a short form of decision

Reviewer’s comment: I also do not understand the purpose for the chi-square given that you are reporting descriptives rather than comparisons

Response: I have removed the chi-square in that I am reporting descriptives rather than comparisons.

Reviewer’s comment: I don’t think H1 is useful or informative. H2 is, but thus needs to be a correlation rather than a regression, this being an exploratory study. In fact, why not just use an RQ?

Response: I have run a correlation analysis in place of H2 and have removed H1 since it is less informative.

Reviewer’s comment: The discussion compared regions, while the RQs and data do not. This is useful info, so I suggest the author, instead of doing the hypothesis, query about regional differences I instead. This can be justified as “delving deeper into the data” so that no changes are done in the from matter. This is assuming that the author addressed the “why” of using the regions in the method section.

Response: I did not compare regions in the discussion. It is not the goal of the study

Reviewers comment: Also, I still do not see how the data is tied to the theory and the three levels of influence

Response: The data was gathered based on knowledge, perception, use and challenges, the theoretical framework offers five analytical lenses through which one can interrogate journalists’ knowledge and use of FOI Act. I made sense of the data using the aforementioned framework.

Reviewer’s comment: Because this is such an exploratory study, the authors may simply dump the theory and examine FOI practices as is.

Response: Based on the response above, I decided to keep the theory section

Reviewer 2 Report

Comments and Suggestions for Authors

Comments on the Quality of English Language

Author Response

Response to Reviewer 2

Reviewer’s comment: Inconsistent Research Questions: The author(s) stated the following as the RQs: What is the level of the knowledge of FOI Act among Nigerian journalists?, How do Nigerian journalists perceive the FOI Act as a useful journalistic tool?, What circumstances and to what extent do Nigerian journalists use the Act in their journalism practice? And What are the challenges faced by Nigerian journalists to the application of the FOI Act in journalism practice? These four RQs were stated on page 6. However, in the result section, the author(s) stated something else. See RQ 3 as stated on page 9. This mismatch needs to be justified.

Response: The inconsistent research question has been vigorously addressed.

Reviewer’s comment: The author(s) should go through their manuscript one more time and check for clarity of argument and coherence.

Response: I have gone through the manuscript and checked for clarity of argument and coherence

Round 2

Reviewer 1 Report

Comments and Suggestions for Authors

The author has addressed a majority of my concerns and that is commendable. The only revision I suggest before publication is to get rid of the Literature Heading and replace it as suggested below:

Replace literature review subheading with a composite subheading such as FoIA in Journalism Practice. This is a better fit for the related subheadings about knowledge, perception, challenges, etc.,  Besides, having a literature review title in the middle of the paper reads awkwardly.

Author Response

I have changed literature review with FoIA in Journalism Practice